# Economic and Environmental Efficiencies of Organizations: Role of Technological Advancements and Circular Economy Practices

Syed Khusro Chishty





Department of Business Administration, College of Administrative and Financial Sciences, Saudi Electronic University, Jeddah 23442, Saudi Arabia; s.chishty@seu.edu.sa

**Abstract:** There are two concepts which encompass the new business order worldwide; this has vast implications, especially in the Indian business scenario. The first blanket is the phenomena of digitalization which was present in the undercurrents of all the business activities from the past two decades, and second one is circular economy practices. But in today's Indian scenario, after the popular reform of "Notebandi", forcing digitalization of the currency puts it to the forefront of all economic activities, especially in India. The economic reform of demonetization highlighted digitalization of economic transactions in the public eye. The phenomenon of digitalization is commonly referred to as the bundle of novel technologies that aim to improve things constantly. Organizations must take advantage of emerging technology to ensure that operations are both economically and environmentally feasible. Technologies based on sustainable solutions might aid companies in becoming more sustainable and economical. Therefore, this research is derived through the desire to measure the economic and environmental performance and how they were influenced by technology and circular practices of Indian Fast Moving Consumer Goods (FMCG) companies as they are most suited for this research. The sample (n = 203) was derived from senior managers of these companies. The study utilized Structural Equation Modeling (SEM) approach to analyze the results, it was found that digital transformation and circular economy practices are pervasive in nature and influence both Economic and Environmental performance of Indian FMCG companies. One of the main contributions of the study is that it also examines the relationship between technological advancements and economic and environmental performance directly; to date, there is not a single study, to the author's knowledge, to have studied this relationship.

**Keywords:** circular economy; Industry 4.0; green economy; SEM; Indian FMCG industry; supply chain; economic performance

## 1. Introduction

Industrial modernization has brought forth detrimental impacts on the ecosystem, encompassing carbon emissions, hazardous chemical leaks, and pollution. Among the various methods devised to foster economic development and sustainable performance, Circular Economy (CE) practices stand out [1]. In a circular economy, the emphasis lies in preserving the value of products and materials for as long as possible. This entails minimizing waste and resource consumption; ensuring resources remain within the economy even after a product's lifecycle ends.

Firms orchestrate and synchronize organizational functions such as marketing, sales, production, logistics, IT, finance, and customer service, both within and across firms. This closure of material and energy loops reduces resource inputs, waste, and emission leakage, subsequently bolstering firm performance [2]. Simultaneously, an escalating awareness of the ecological impacts of manufacturing is compelling producers in emerging and developed markets alike to reconsider their practices. However, the pressure for producers to contribute continuously to their countries' economic development is also mounting. This

necessitates a delicate balance between commercial growth and environmental preservation, given the predicted rapid expansion of the manufacturing industry over the next decade [3]. Consequently, manufacturers are recognizing the urgent requirement to adopt environmentally conscious approaches, including recycling initiatives, in collaboration with consumers and suppliers, to mitigate the negative ecological consequences of their products and services [4]. The topic of the CE is drawing increasing attention from industries, academics, policy makers, and consumers [5]. In particular, there is a pronounced need for academic research to ascertain whether CE activities yield the desired firm performance outcomes.

Concurrently, contemporary technologies play a pivotal role in facilitating Circular Economy (CE) practices, with this study advocating for the significance of technological advancements, such as blockchain, in supporting organizational performance paradigms [6]. These advancements are recognized as crucial enablers capable of addressing diverse firm performance challenges [7]. Advanced technologies (AT) have the potential to drive economic transitions by amalgamating, analyzing, and integrating data [8]. For example, the establishment of digital networks to streamline information flow in supply chain processes holds substantial potential for waste reduction, enhancing circular resource flows, and refining decision-making [9]. Consequently, there is a pressing need to scrutinize the role played by advanced technologies, including big data, AI, and blockchain, in achieving sustainability within business operations.

Both CE and AT signify emerging concepts capable of instigating socioeconomic transformations [10]. CE's central goal is to elevate product value by extending their lifespans, necessitating a reshaping of business models and production systems during the transition from linear to circular economies [11]. The confluence of AT and the CE model in augmenting firm economic and environmental performance has captivated researchers in the nascent fields of digitalization and sustainability [12]. Clarifying these phenomena can yield a better comprehension and implementation of sustainable business models. Consequently, we formulate the following research question:

> *What relationship exists between advanced technology, the circular economy, and the environmental and economic performance of firms?*

A plethora of research on practices and firm performance has emerged since its inception [13]. Empirical research outcomes on the impact of CE practices on enterprise performance vary. For instance, refs. [14,15] found that the implementation of CE practices did not lead to financial growth for Chinese industrial enterprises. The infancy of CE practice implementation implied substantial investment costs, potentially increasing operating costs and subsequently decreasing commercial profits. However, more recent research has suggested a positive correlation between CE practices and financial benefits [16]. Given these mixed results, our research seeks to offer deeper insights into the correlation between CE practices and enterprise performance, recognizing the intricate and multifaceted nature of CE implementation.

In this study, we employ regression analysis to explore the relationship between CE, AT, and their impact on the economic and environmental performance of Indian FMCG firms. As the Indian FMCG sector presents a unique context, we propose that digitalization holds substantial potential to enhance the sector's transition to a circular economy. The scientific literature and public discourse underscore digital technologies' role in promoting circular economies—economies that minimize resource use while maximizing material value through practices such as reuse, repair, refurbishment, remanufacturing, recycling, and energy recovery [17,18]. Industry 4.0 technologies, including IoT, big data, advanced analytics, 3D printing, additive manufacturing, blockchain, and online platforms, are expected to drive circularity in business models, products, and production processes. They also facilitate knowledge exchange and connections between stakeholders across the value chain [19]. Additionally, digital technologies are poised to empower citizens and consumers by providing information and education, and encouraging active participation in the transition towards circular economies.

Existing research on the advanced technology and circular economy has largely focused on investigating technologies' enabling role in circular strategies [10,20], identifying use cases [21–23], and addressing implementation challenges [24]. However, the current literature lacks a comprehensive analysis of the opportunities and risks posed by AT in achieving environmental, and economic sustainability [4]. Consequently, there is a growing call for more research on the impact of digital technology and circular economy within a specific economy context [25–28]. This article endeavors to bridge this knowledge gap by offering an investigation into the effects of AT and CE on the environmental and economic performance of the resource intensive FMCG sector within a developing economy such as India, particularly in a resource-intensive sector.

## 1.1. FMCG Sector

The FMCG (Fast-Moving Consumer Goods) sector, the fourth-largest contributor to the Indian economy, is on a remarkable growth trajectory. As of December 2022, the FMCG market had already surged to a valuation of US$ 56.8 billion. Projections foresee a remarkable journey ahead, with the total revenue of the FMCG market poised to exhibit a robust Compound Annual Growth Rate (CAGR) of 27.9% between 2021 and 2027, potentially catapulting to nearly US$ 615.87 billion by 2027 [24]. What is particularly noteworthy is the sector's resolute commitment to sustainability and circular economy practices. A noteworthy 60% of FMCG companies in India have embraced at least one circular economy practice, as highlighted in the FICCI Report of 2022. This dedication extends further, as a survey by the Confederation of Indian Industry (CII) revealed that an impressive 75% of FMCG companies operating in India are actively engaged in sustainability initiatives [29,30]. Aligned with these sustainability objectives, the Indian government has set an ambitious target to reduce waste generation by a substantial 30% by the year 2030 [24]. To foster green manufacturing, a series of regulations have been introduced, including the Energy Conservation Act and the Water Conservation Act. FMCG companies are responding to these imperatives by integrating sustainable ingredients and materials into their product portfolios. For instance, Hindustan Unilever has committed to incorporating 100% sustainable palm oil in its products by 2023, while Nestle India is diligently working towards using 100% recycled paper in its packaging by 2025 and enhancing packaging recyclability [31]. The tangible expression of this commitment can be observed in Coca-Cola India's innovative introduction of a bottle crafted entirely from 100% recycled plastic and PepsiCo India's launch of snacks thoughtfully packaged in compostable wrappers. Moreover, these companies are proactively conserving water within their operations; Dabur India, for instance, has deployed water-saving devices in its manufacturing facilities, and Godrej Consumer Products has introduced a rainwater harvesting system at its corporate headquarters.

Beyond this, significant strides are being made in enhancing energy efficiency, as exemplified by ITC's installation of solar panels at its factories and Marico's implementation of energy-efficient lighting within its offices. Additionally, FMCG companies are placing an emphasis on responsible waste management to minimize their environmental footprint. Notable initiatives include Procter & Gamble India's establishment of a waste-to-energy plant at its Manesar factory and HUL's pioneering recycling program for its shampoo bottles [32].

These noteworthy sustainability milestones underscore the substantial progress achieved by FMCG companies in India in their unwavering commitment to sustainability. Therefore, the choice of the FMCG sector as the focal point of our study is well-justified, as it embodies the transformative journey toward sustainability and green manufacturing.

## 1.2. Review of Literature

### 1.2.1. Economic and Environmental Performance

The central focus of organizations revolves around maintaining both economic and environmental efficiency in their operations, with the aim of flourishing and competing effectively in the marketplace. The achievement of organizational performance is con-

tingent on a multitude of factors, encompassing the optimization of economic gains, the empowerment of human resources, the cultivation of innovative expertise within the organization, and the advancement of environmentally responsible operational practices. This concept of sustainability transcends mere financial metrics such as profits and return on investment; it encompasses a holistic approach that takes into account environmental and social dimensions, as elucidated by [33,34]. Manufacturing enterprises are tasked with fulfilling the expectations and requirements of diverse stakeholders such as clients, customers, suppliers, society, and governments. To attain the pinnacle of sustainability, manufacturing entities must address two interconnected dimensions: the economic dimension and the environmental dimension, as coined by [35,36]. The economic sustainability performance (EP) holds a pivotal role in ensuring the financial well-being of a company. The ability to continuously produce goods and services while simultaneously generating profits is quintessential for the organization's survival, a principle emphasized by [5]. The sustenance of sound economic performance is a critical gauge of a business's capacity to thrive in the long term, and it greatly influences a wide array of decision-making processes that reverberate across various aspects of the organization, as underscored by [36]. The environmental sustainability performance (EVP) centers on an organization's ecological impact arising from its routine production activities, as elucidated by [5,37]. Thus, to ensure environmental sustainability, an organization must endeavor to achieve either net zero emissions or a positive ecological footprint within its local ecosystem. This involves endeavors such as improving air and water quality, harnessing local waste streams, deploying renewable energy sources, and functioning as a reservoir for surplus energy, as posted by [38].

### 1.2.2. Circular Economy

In alignment with other scholars [10,26,39], we contend that the prevailing linear economic model, characterized by inefficient resource utilization, serves as the fundamental catalyst for numerous prevailing environmental crises. These encompass issues such as the depletion of natural habitats, scarcities in resources, unsustainable production–consumption waste, oceanic plastic pollution, and escalating health concerns arising from burgeoning waste volumes [25,29,40]. As these interconnected challenges propagate across the globe via intricate supply chains, the linear economy (LE) emerges as a formidable "societal grand challenge", a term defined by [41] to denote global predicaments impacting vast populations across multiple nations and regions.

In response to the impending consequences of linear production–consumption patterns, a prevalent solution advocated is the transition toward a circular economy (CE). This novel economic paradigm revolves around business models that transcend the traditional "end-of-life" concept by prioritizing reduction, reuse, recycling, and resource recovery throughout production, distribution, and consumption processes. Its application spans micro-level (products, companies, consumers), meso-level (eco-industrial parks), and macro-level (city, region, nation, and beyond), with the ultimate objective of fostering sustainable development—entailing environmental integrity, economic prosperity, and social equity—for the present and future generations [6,31,42]. The burgeoning interest in CE is mirrored by the surging volume of recent investigations. Initial studies in this domain [33,43,44] concentrated on framing the CE concept and distinguishing it from related notions such as sustainability. More contemporary inquiries have zoomed in on specific aspects, including CE business models [16,45], circular supply chains [8], digital technologies in CE [21,42,46,47], and geographical analyses of CE implementation [35,37]. Consistently recurring themes in the existing research spotlight various conceptualizations of CE. Some scholars view it as an overarching economic system [48] or as a mechanism for waste elimination, while others perceive it as a holistic vision for societal transformation [49]. Conversely, CE is also construed as an assembly of strategies [49] or as a targeted solution to specific issues [50]. Further conceptualizations encompass CE as a means to decouple economic growth from resource usage [16,44,51], a mechanism to uphold material

values, or a pathway to regeneration [52,53]. Despite the considerable scholarly attention lavished on this topic, a notable dearth of critical exploration persists in the domain of organizational performance concerning the influence of CE practices on economic and environmental performance. Noteworthy investigations [5] that have linked CE to organizational performance have yielded mixed findings. For instance, authors of [46] report a positive correlation between the adoption of green practices and firms' economic value. Similarly, some studies posit a favorable association between green supply chain practices and firms' profitability [1]. Conversely, counter-evidence suggests that the embrace of green practices could lead to reduced productivity and financial pressures [53,54]. This incongruity underscores the need for an in-depth exploration of the role of CE practices in enhancing organizational performance. Researchers contend that contextual factors play a pivotal role in elucidating the multifaceted outcomes observed in CE-performance research. Thus, our investigation focuses on a developing economic context within the resource-intensive FMCG sector.

As highlighted by [55], Circular Economy (CE) holds a pivotal role in ensuring economic security, fostering green growth, and promoting sustainable economic development. This concept not only carries substantial potential to generate novel and unparalleled opportunities but also underscores the recommendation for nations to implement dedicated policies in support of Circular Economy principles [26,56]. This notion advocates circular economy practices plays a significant role in producing sustainable solutions. The concept of technological advancement and circular economy is a new research interest and needs to be explored.

### 1.3. Theoretical Framework and Hypothesis Development

#### 1.3.1. Conceptual Model

This research proposed a conceptual model containing all the research variables. The conceptual model is given as under (Figure 1).

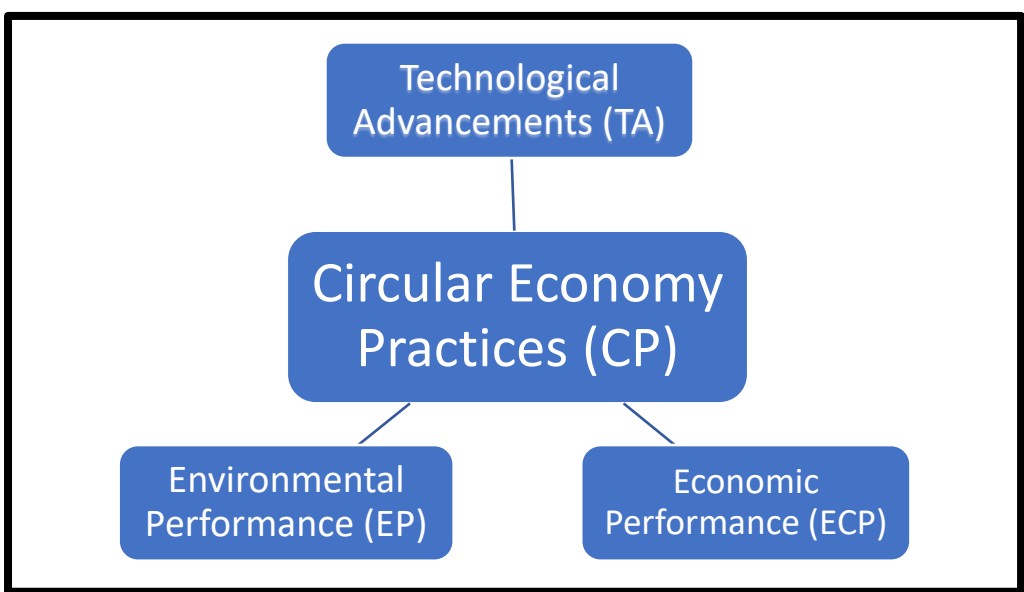

**Figure 1.** Conceptual Frame of the Study.

#### 1.3.2. Technology Advancements (TA) and the Circular Economy Practices (CP)

This section is designed to explore the relationship between technology integration and green practices or circular economy practices. Presently, digitalization has generally been considered as an essential element in every industry and has quite an impressive role in the evolution of the CE industry. These systems may be used to decrease the consumption of resources. It can be used lower logistics cost. Technology innovation

encourages transparency, which enables the company to obtain information on the usage of resources for the product, assisting businesses in extending the life of their products and moving towards CE practices.

Digital technologies utilized in Industry today have been categorized by [8,57] into three streams of data management: collection of data, data processing, and data analysis. Data-gathering technologies include sensors (such as RFID) and devices that connect objects and people to the Internet. Data integration technologies gather and categorize data, and data analysis prepares data for decision making [23]. The most often mentioned technologies are blockchain and the cloud [46]. Furthermore, analyzing big data and artificial intelligence (AI) are now widely used as essential tools for analyzing data. Data are becoming a fifth factor of production, the companies dealing in this area are precious. The use of data is pervasive and essential in every field of human enquiry. Therefore, circular economy practices also suffice from it and gets heavily influenced.

According to researchers, CE models seek to build waste-reducing goods and services with the use of digital technology in order to promote sustainability [58,59]. Existing research has shown that the status of digital technology is having a growing impact on CE practices and innovation [19,39]. Additionally, efforts are being made to integrate digital technology into the industrial sector at the same time as CE development [7]. In addition, it has been stated that cutting-edge digital technologies are swiftly taking hold in industrial transformation, including the Internet of Things (IoT), RFID, Internet-based Services (IoS), and meta cyber systems (MCS) [20]. The retail industry and the downstream portion of the upstream supply chain depend heavily on the data generated by technologies such as RFID. Companies may evaluate the quality of returned items using these data, and return flows can be optimized over the course of the PLC [1]. The most recent research by [18,27] discovered that businesses that implement circular economic concepts experience much higher value generation from technological development. Therefore, based on the aforementioned justifications, it may be concluded that:

**H1.** *Circular Economy practices are greatly influenced by technological innovation and advancements.*

### 1.3.3. Technological Advancements and Environmental Performance

Using Internet also showed how real-time data for decision-making may assist cyber-physical systems optimize production, supply, and maintenance [10]. Incorporating the aforementioned technology solutions into material processes also aids in the collection, organization, and use of trash as a resource for the company [28,30].

**H2.** *Environmental Performance is greatly influenced by technological innovation and advancements.*

### 1.3.4. Technological Advancements and Economic Performance

According to studies on Industry 4.0, digital technologies also help businesses become efficient by cutting expenditure by utilizing capabilities of IT. Digital technologies not only assist companies in enhancing value generation and capture, but also influence resource flow plans. Digital solutions' ability to automate resource management, control, and optimization may be used to develop circular business models, which also helps firms reduce supply chain costs.

**H3.** *Economic Performance is greatly influenced by technological innovation and advancements.*

### 1.3.5. Circular Economy Practices (CP) and Environmental Performance (EP)

Circular Economy Practices such as procurement or design are two environmental projects that are frequently recognized as good strategies to significantly minimize waste output and decrease the overall ecological footprint [1]. The future of supply chains using digital technologies and circular economy principles asserts that the introduction of CP is a

crucial first step in the development of a fully green supply chain process. Environmentally friendly design and operating practices significantly reduce harmful environmental effects and may increase the viability of the business. The ecological design makes it easier to disassemble and recycle items, which will enable the company to use fewer dangerous chemicals and use less raw resources in manufacturing. When refs. [47,60] looked at several environmental performance factors, they also discovered that eco design is strongly and directly connected to socio-environmental sustainability.

The researchers contend that selecting the appropriate suppliers is just as crucial for businesses as adopting sustainable green management [26]. Having an extraordinarily wide selection of environmentally friendly providers is crucial in a difficult, cutthroat market, impacting production choices both fundamentally and psychologically. In their study of UK businesses, ref. [12] found a significant and positive correlation between eco-practices and environmental performance. By implementing green practices at various points throughout the supply chain, businesses may be able to increase income while reducing waste and improving processing efficiency. Every attempt is made to reduce the unfavorable environmental effects of a company's products and services as part of its GSCM operations. According to [26], these initiatives help reduce material and water use, as well as trash production, to the absolute minimum. Therefore, based on the aforementioned justifications, it may be concluded that:

**H4.** *The Circular economy Practices influence positively to Environmental Performance.*

1.3.6. Circular Economy Practices (CP) and Economic Performance (ECP)

Organizations benefit economically from implementing technology-based systems to advance the circular economy in numerous ways, some of which were already discussed in the introduction section. According to the growing body of research on the circular economy, the inclusion of technology may lower long-term costs related to environmental risks associated with corporate operations [1]. The use of CE-accredited methods helps to conserve natural resources, reduce wasteful spending, and keep money in the general economy [47]. Eco-friendly practices assist in the reduction of waste throughout whole operations, increasing profits [45]. According to the literature, green practices and business performance are related. Two methods in which GSCM may improve economic performance are as follows: first, enterprises can gain an economic advantage by using less resources and energy. Second, businesses may obtain indirect economic benefits by improving their brand recognition and customer loyalty [61]. According to these studies' findings [24,27], applying Green SCM approaches positively affects a company's total productivity. Additionally, market-driven circular economy initiatives such as creating more sustainable products may expand sales opportunities and increase profitability [52,53]. The research by [62], also advocated the similar perception, validated this earlier research. Green management practices have been shown to improve public perception and reputation, which might lead to increased demand for goods [21,63]. Increasing operational efficiency through waste reduction is also thought to improve environmental performance, which, in turn, leads to better financial performance [64]. As a result, it is implied that based on the reason and previous arguments.

**H5.** *The Circular economy Practices influence positively to Economic Performance.*

## 2. Materials and Methods

### 2.1. Sampling Procedure

A digital survey was employed in this study to gather data. The replies were received from industrial companies in India. This cross-sectional research designed study utilizes convenience sampling, in which 300 questionnaires were distributed to various automotive industry companies via emails, and other forms of contact. Of the 229 replies, 26 were

deemed invalid and were not included in the study's sample. Thus, the item completion rate was quite impressive, at more than 90%. The remaining 203 questionnaire responses, representing a response rate of 53.25 percent, were analyzed in order to evaluate the hypotheses. The general idea behind the current investigation is shown in the conceptual model (Refer Figure 1).

The study procured a list of respondents through the ranking list published by Business Standard, in 2022. The responding organizations were utilized in two steps:

**Step 1.** The list published in business Standard 2022 and contains Indian and foreign companies. Based on their country of origin, only Indian companies were shortlisted. After the selection, 945 Indian companies were chosen for the next step.

**Step 2.** From this frame of 945 organizations, FMCG companies were identified. This was essential as the study was focused on supply chain operations of FMCG companies. In total, 300 FMCG companies were finally selected for the study.

The companies listed in this magazine were all within the top most listed companies having structured and well-defined practices. The managers working in them are considered as subject matter experts due to their huge experience in the field. Therefore, these respondents and responding organizations were chosen for the study.

### 2.1.1. Survey Instrument Development

We started by creating a questionnaire, which we then pre-tested with specialists (five academics and five supply chain professionals). As a result, certain adjustments were made to the measures to make sure that the language was precise and understandable. The 5-point Likert's questionnaire was followed by a covering letter.

Summary and definition of research constructs are presented in Tables 1 and 2.

**Table 1.** Summary of research constructs.

| S.No | Construct | Author |
|------|-----------|--------|
| 1 | TA | [49,65] |
| 2 | CP | [15,53] |
| 3 | EP | [38] |
| 4 | ECP | [3] |

**Table 2.** Definition of research constructs.

| S.No | Construct | Definitions |
|------|-----------|-------------|
| 1 | TA | Technological advancements include use of internet-based systems e.g., blockchain, RFIDs, Self-Check out big data, and artificial intelligence. These intelligent technologies assist businesses in enhanced supply chain operations to follow circular economy principles [49,65]. |
| 2 | CP | CP place a strong emphasis on working with suppliers to use environmentally friendly materials that are easily recyclable and remanufactured and follow green practices in supplying them [31,60]. |
| 3 | EP | It refers to businesses' capacity to safeguard the environment by cutting back on waste, energy use, and harmful chemicals along the whole supply chain [3,38]. |
| 4 | ECP | According to [3] ECP refers to a production facility's ability to lower the prices of materials and component supply, recycling and remanufacturing processes, and waste disposal. |

### 2.1.2. Method of Analysis and Sample Adequacy

The best statistical method for assessing hypotheses and structural models based on questionnaire responses is structural equation modelling (SEM). SEM, a second-generation method, is capable of examining the intricate connections between latent components. In analyzing the current study model, Covariance-Based Structural Equation Modelling (CB-SEM) was employed. This method is a popular method of analysis in this domain used by many researchers [14,47]. In general, scientists concurred that CB-SEM is better suited for evaluating hypotheses. Additionally, it accurately evaluates the covariance matrix and offers many indices, such as GFI, CFI, and RMSEA, which can aid in determining the model's level of fitness (Table 3).

**Table 3.** Descriptive Statistics.

| Demographic Variables | Frequency | Percent |
|---|---|---|
| Manager | 158 | 77.8 |
| Senior Manager | 26 | 12.8 |
| Vice president | 6 | 3.8 |
| CEO | 13 | 6.4 |
| **Total** | **203** | **100.0** |
| **Experience (Present Position)** | | |
| 0–5 years | 118 | 58.0 |
| 5–10 years | 15 | 13.0 |
| 10–15 years | 40 | 19.0 |
| More than 15 years | 30 | 14.0 |
| **Total** | **203** | **100.0** |
| **Total Experience** | | |
| 0–10 years | 97 | 47.9 |
| 10–20 years | 103 | 50.9 |
| 20–30 years | 3 | 1.5 |
| **Total** | **203** | **100.0** |
| **No of employees** | | |
| 250–500 | 119 | 59.0 |
| 500–1000 | 71 | 34.0 |
| More than 1000 | 13 | 6.4 |
| **Total** | **203** | **100** |

### 2.2. Response Bias and Non-Response Bias Assessment

Before beginning the data analysis, the data's response and non-response bias was evaluated.

### 2.2.1. Response Bias Assessment

Response bias can steer participant replies in self-reporting research away from the ideal response. Additionally, it undermines the survey instrument's validity. Steps were recommended by [11,14,66,67] to lessen response bias in self-reporting studies. A certain number of these were used when creating a questionnaire. For example, the respondent's identity and the nature of his or her replies were kept private. Responses will only be used for research purposes; this has been guaranteed. Additionally, it was made clear that only overall replies would be analyzed and that the researcher would not track down and make use of individual responses. To obtain accurate replies, the appropriate labels were placed

in front of the items. Additionally, efforts were taken to eliminate biases identified by [4,13] by maintaining the wording of the scale as straightforward and plain as possible.

### 2.2.2. Non-Response Bias

This bias developed as a result of some respondents responding slowly or not at all. To examine the differences between respondents (early) and late or non-responders, independent sample t-tests can be used. If no differences were found, as stated by [2], response bias does not exist.

Any potential for non-response bias may be evaluated using the demographic profile of the respondents (e.g., designation, experience) [2]. The information was gathered from managers at various levels of the organizational structure, which is depicted in Table 4. The sample representation of respondents and the normality of the data were also evaluated and determined to be appropriate. Therefore, non-response bias is absent in this research.

**Table 4.** Group Statistics for Estimation of Non-Response Bias.

|  | **Constructs** | **N** | **Mean** | **Std. Deviation** | **Std. Error Mean** |
|---|---|---|---|---|---|
| TA | Early | 183 | 3.4455 | 1.32250 | 0.28173 |
|  | Late | 20 | 2.0645 | 1.32269 | 0.24110 |
| CP | Early | 183 | 2.8636 | 1.35260 | 0.28102 |
|  | Late | 20 | 1.2581 | 1.12251 | 0.20197 |
| EP | Early | 183 | 2.6364 | 1.39282 | 0.29123 |
|  | Late | 20 | 1.7097 | 1.48258 | 0.26118 |
| ECP | Early | 183 | 1.6455 | 1.32250 | 0.22173 |
|  | Late | 20 | 2.1645 | 1.33269 | 0.23110 |

To assess normality of data Table 5, a $z$-test is applied as a normality test using skewness and kurtosis. A $z$ score could be obtained by dividing the skewness values or excess kurtosis value by their standard errors. For small sample size ($n < 50$), $z$ value $\pm 1.96$ are sufficient to establish normality of the data. However, medium-sized samples ($50 \leq n < 300$), at absolute $z$-value $\pm 3.29$, conclude the distribution of the sample is normal [68]. In this research $z$ statistic is greater than 3.50 for all research parameters ensuring normality of the data.

**Table 5.** Showing basic descriptive statistics for normality of data.

| **Construct** | **Mean** | **Std. Deviation** | **$z$ Value** |
|---|---|---|---|
| TA | 4.566 | 1.34 | 4.45 |
| CP | 5.434 | 1.08 | 3.89 |
| EP | 4.555 | 1.56 | 5.76 |
| ECP | 4.098 | 1.24 | 6.04 |

### 2.2.3. Common Method Bias

Common method bias is the bias introduced by the scale as a result of some external problem, often at the time or way of collecting data using a single (common) technique. When creating and constructing the scale, procedural approaches can be implemented into the research instrument. The following are some of the procedural techniques:

(1)   Maintaining the anonymity of respondents

As indicated, responses from the responder were kept private and confidential. Additionally, it was guaranteed that replies would only be utilized for research. A covering letter

with a confidentiality and response-utilization clause was included with the questionnaire to address these concerns.

(2)  Rearranging scales and enhancing scale components

Independent variables are placed before independent variables when using the scale reordering approach [4]. This approach lessens typical technique bias as well.

(3)  Statistical Procedures

When most of the variance can be accounted for by one component, a study is said to have considerable common method bias. To determine whether there was a possibility of [4] suggestion of common technique bias, Harman's single factor test was used. The Harman single factor test may be used to calculate the variation explained by a single factor. On all research scales, exploratory factor analysis (EFA) was carried out. A common method bias exists when the majority of the variation is explained by a single component. Eight variables were developed in the current study.

### 2.3. Data Analysis Technique

More than 300 respondents were contacted. Of the 229 surveys, 26 were incomplete and were not used for further research, leaving 203 responses as a final tally for analysis. When using SEM to provide significant findings, sample size is crucial. For moving further with SEM utilizing Maximum Likelihood Estimation (MLE), a sample size of 100–200 is often advised. However, researchers contends that when considering maximum likelihood estimate, sample sizes of 100 are strong. With three or more indicators per component, a sample size of more than 100 will often be adequate for convergence and a good solution [11].

For each free parameter assessed, 10 participants are said to be adequate. This well-known rule of 10 is referenced in a number of articles that are regularly quoted [40,41]. SEM may be used securely because the current study comprises only 4 free parameters and a sample of 203 responses. The following formula was also provided by Joreskog and Sorbom [35] to determine the minimal sample size:

$$k\,(k-1)/2,$$

where $k$ is the number of variables.

## 3. Results

### 3.1. Measurement Model Assessment

Measurement model was assessed for all research constructs (Table 6). Before proceeding with the structural model, assessment of the measurement model is must. This is a preliminary step in structure equation modelling.

**Table 6.** Showing items of all research constructs.

| Nature of Constructs | Variables | Items (Original) | Items (Refined) |
|---|---|---|---|
| Independent Variables | 1. Technological Advancement (TA) | 7 items | 3-item Scale |
| | 2. Circular Economy Practices (CP) | 9 items | 3-item Scale |
| Dependent Variables | 1. Economic Performance (ECP) | 9 items | 3-item Scale |
| | 2. Environmental performance (EP) | 6 items | 4-item Scale |

#### 3.1.1. Testing Unidimensionality

It is crucial to examine one-dimensionality, and when conducting the evaluation, CFA was used to make sure that no item should load on any other element. According to [69], CFA is thought to be the best way for ensuring unidimensionality out of all available techniques.

Confirmatory Factor Analysis (CFA)

CFA is used to evaluate model fit, which must be within an acceptable range before moving on with a structural model, as well as reliability, validity, and fit to the data. Fit indices are crucial for evaluating the measurement model for the study (Table 7). The disparities between the observed and calculated covariance matrices among the Scale items are known as fit measures, which also guarantee that the data accurately represents the underlying theory (Figure 2).

**Table 7.** Fit Indices.

| Constructs | Fit Indices | | | | | | |
|---|---|---|---|---|---|---|---|
| | GFI | AGFI | NFI | NNFI | CFI | SRMR | RMSEA |
| TA | 0.92 | 0.81 | 0.88 | 0.83 | 0.88 | 0.06 | 0.11 |
| CP | 0.91 | 0.89 | 0.95 | 0.84 | 0.92 | 0.04 | 0.07 |
| EP | 0.97 | 0.92 | 0.92 | 0.88 | 0.93 | 0.04 | 0.10 |
| ECP | 0.94 | 0.88 | 0.94 | 0.94 | 0.90 | 0.05 | 0.08 |

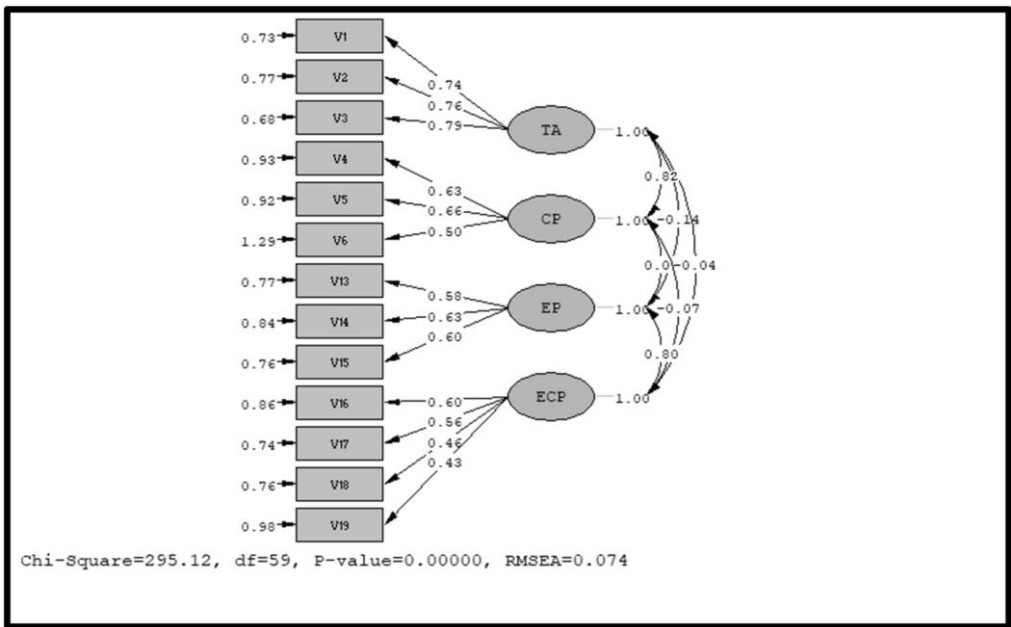

**Figure 2.** CFA of Study Scales.

Since the values vary on the sample size, it is challenging to obtain precise fit indices for all indices. Fit indices cannot, therefore, be used to distinguish between excellent and bad models. Since different measures present different aspects of the model, it has been difficult to agree on a single measure up to this point.

The most common fit indices utilized in management science research are CFI, GFI, NFI, and NNFI. CFI and RMSEA are the most often used fit measures since they are independent of sample size [67].

The CFA was conducted using all the research scales, and a measurement model was created. When it was discovered that some of the standard loadings were outside of permissible bounds, the measurement model was rerun until all of the standard loadings were suitable. Finally, a nine-item scale was determined to be unidimensional.

3.1.2. Reliability Evaluation

After determining unidimensionality, a reliability analysis of all research measures was completed before moving on to the assessment of the validity [55]. It has been suggested

that dependability/reliability must first be established before validity can be evaluated. Scale reliability's internal consistency and stability were evaluated in order to gauge the reliability of scales.

Scale Reliability

Scale reliability is the capacity of a scale to assess the scale Consistency. In addition to being a need for validity, it is also strongly tied to it [44,54]. The following criteria can be used to evaluate it: Cronbach's alpha is the most popular and widely used approach for determining dependability. Ref. [70] suggested values for this measure should range from 0.6 to 1.0. Cronbach alpha is shown separately for each research scale in Table 8. The reliability values for each scale are shown in Table 8. Scales are trustworthy because all values are acceptable.

**Table 8.** Cronbach 's alpha, CR, and VE.

| Constructs | Cronbach's Alpha | Constructs Reliability (CR) | Variance Extracted (VE) |
|:---:|:---:|:---:|:---:|
| TA | 0.624 | 0.8 | 0.5 |
| CP | 0.625 | 0.7 | 0.5 |
| EP | 0.611 | 0.7 | 0.4 |
| ECP | 0.638 | 0.7 | 0.5 |

It is proposed that in order to demonstrate a high level of dependability, the values must be more than 0.5. All results for the current study, including Cronbach's alpha and construct reliability, fall within an acceptable range, indicating dependability. The VE values also meet acceptable limits.

*3.2. Validity Evaluation*

The validity of a research scale is determined by the scale's capacity to assess the variables it is intended to evaluate properly. All of the research scales underwent several types of validities assessments. The construct validity of the practical tests and items derived from the literature is their ability to measure what the theory claims. It entails providing the theory relating to study topics with theoretical and empirical justification. Additionally, it includes statistical analysis of test internal design and the connections between answers to various test components. Additionally, they provide connections between test results and measurements of other components. Convergent, discriminant, and nomological construct validity were the three categories that were evaluated. It demonstrates how closely related the objects on a single scale are to one another. According to [66], a scale is considered to be convergent when elements of a certain concept share a significant amount of variation. There are several methods available to calculate the convergent validity:

High standard loadings can be used to establish it [32,66]. To prove convergent validity, additional measurements are also taken. For instance, NFI and NNFI values greater than 0.9 indicate that the scale is validly convergent [71]. T-value findings were also recommended by [55] for proving convergent validity. For the study scale, the *t* values should be greater than 2. All the research scales in the present research had *t* values greater than 2, which is a sign of excellent convergent validity. The convergent validity values for each scale are displayed in Table 9.

**Table 9.** Loading Values, NFI, NNFI, and T-Values for Convergent Validity.

| Constructs | Loading Value | NFI | Range of *t*-Values |
|:---:|:---:|:---:|:---:|
| | Range | | |
| TA | 0.74–0.79 | 0.90 | 5.75–18.81 |
| CP | 0.50–0.63 | 0.91 | 11.64–14.16 |
| EP | 0.58–0.60 | 0.92 | 10.26–13.16 |
| ECP | 0.43–0.60 | 0.90 | 8.76–12.21 |

*3.3. Structural Model Assessment*

All research constructs were projected into a single model. The direct relationship between constructs was assessed using the direct effect model. It is advised that if the comprehensive model does not converge, separate models can be assessed. In this study, all scales were made unidimensional, and a thorough model that included all independent and dependent variables converged with fit indices. Figure 3 depicts the model. The independent variable (X) and dependent variable (Y) should be associated in the structural model. The assumptions were evaluated using the route coefficients of the direct effect model. The direct impact structural model is shown in Exhibit 4. Following are the findings of the direct effect model hypothesis testing.

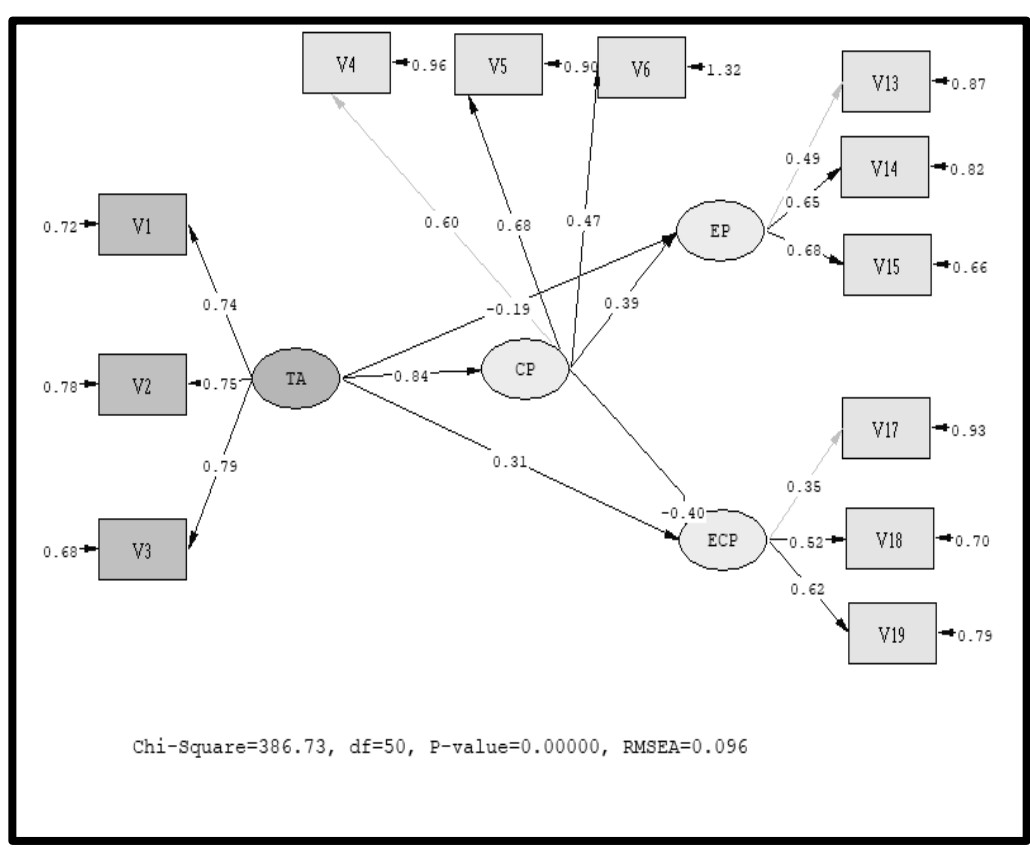

**Figure 3.** The Direct Effect Structural Model.

The results demonstrate a statistically significant, direct, and favorable relationship between technological innovation and organizations' circular economy practices (r = 0.84, *p* = 0.001) As a result, the results support H1 supports to the notion that TI is essential to improving CP for SMEs. In the case of FMCG firms, the findings regarding Technological advancement construct Environmental performance were not significant. However, technological advancement has a strong impact on Economic performance of FMCG companies.

H2, = −0.19, $p$ = 0.001; H3, = 0.31, $p$ = 0.001. Additionally, it is shown that CP aids the firm in achieving environmental efficiencies by supporting the hypothesized relationships between circular practices and environmental performance (H4, = 0.39, $p$ = 0.001), CD and ECP (H5, = −0.40, $p$ = 0.001); therefore, H5 is, however, rejected since there was no statistically significant correlation between CP and ECP. The findings of the route between CP and performance show that CP only significantly influences EP alone. A summary of all the hypotheses is presented in Table 10.

**Table 10.** Summary of Hypotheses testing based on estimates provided in SEM.

| S.NO | Result | Relationship | Estimates |
|------|--------|--------------|-----------|
| H1 | Accepted | TA-------CP | 0.80 |
| H2 | Not Accepted | TA------EP | −0.19 |
| H3 | Accepted | TA-------ECP | 0.31 |
| H4 | Accepted | CP------EP | 0.39 |
| H5 | Not Accepted | CP-------ECP | −0.40 |

## 4. Discussion and Implications

The findings of the present study underscore the significant influence of Advanced Technologies (AT) on Circular Economy (CE) practices. These results align with earlier investigations by [22,47], corroborating the substantial impact of AT on CE practices. These studies also emphasized the challenges of technological security and interoperability faced by this technology. In a similar vein, ref. [22] positioned AT, such as blockchain, as an emerging facilitator of CE practices, supporting information systems and enhancing CE performance. Ref. [72] conducted a case study that reinforced the idea that AT fosters energy and material traceability, streamlining reuse and recycling planning, making it a feasible strategy for CE adoption. Researchers stressed the pivotal role of AT in implementing CE, particularly through digitization, integration, and automation during the current era of industrial revolution.

Remarkably, our findings indicate that the advancement of technology does not significantly impact environmental performance, in accordance with recent work by [7]. Nevertheless, this conclusion contrasts with the study by [19] conducted within the context of Mexican SMEs. This inconsistency may arise from the fact that Indian technologies are not tailored to enhance environmental performance. Alternatively, the impact of AT on environmental performance might be indirect, warranting further investigation. This outcome enriches the literature by suggesting that the direct impact of AT on environmental performance within India's FMCG sector is not prominent.

Additionally, our research underscores the positive influence of AT on the economic performance of Indian FMCG companies, consistent with earlier studies. Moreover, we find that circular economy practices yield a positive impact on the environmental performance of Indian FMCG firms, aligning with existing research [47] However, it is worth noting that our findings do not reveal a significant impact of circular economy practices on economic performance, which diverges from prior studies [47]. This unexpected outcome may arise from two factors. Firstly, Indian FMCG companies might not have fully embraced CE practices to enhance economic efficiency. Secondly, due to the resource-intensive nature of the FMCG sector, its efficiency may be more dependent on resource availability rather than CE practices. This finding contradicts prior conclusions suggesting that CE practices boost economic performance. Consequently, we recommend further research to delve into the underlying causes of this inconclusive outcome.

The study's contributions are manifold. It provides a contextual understanding of CE practices and organizational performance within the framework of a developing economy such as India, addressing a gap in a literature predominantly focused on Western developed nations. Additionally, the intriguing disparity in findings regarding the connection

between CE practices and organizational efficiency is highlighted. Our study highlights that technological advancements primarily impact economic efficiency, while CE practices predominantly influence environmental efficiency. These revelations pave the way for future research and the need for further validation. This study also provides a clearer understanding of FMCG sector of Indian economy which extremely important from CE perspective as it is resource intensive sector. Lastly, our study responds to the call for exploring the nexus between CE practices and firm performance, contributing to the development of a more robust framework and theory-building process.

The goal of the current research is to improve organizational economic and environmental performance by examining the impact of digitalization on CE practices. Through various channels of contact, questionnaires were used to gather cross-sectional data from Indian FMCG Companies. The SEM method was utilized for analysis. The analysis' results were found to be internally consistent and to have both convergent and discriminant validity. The findings demonstrate that digitalization has a favorable impact on CE practices. The results also show that the firm's performance in terms of the environment is significantly improved by CE practices. The study's findings also show a strong relation between technological advancements economic performance. The study also reveals a negligible impact of circular economy practices on economic performance of these firms. The relationship between technological advancements and environmental performance found to be insignificant.

### 4.1. Theoretical Implications

The findings demonstrate that technological advancements have a favorable impact on CE practices. This indicates that technology advancements helped East European automakers accelerate their adoption of CE. The results also show that CE practices are shown to significantly improve a company's operational, economic, and environmental performance making those businesses more sustainable as a consequence. The study's findings also show a strong correlation between technological advancements, CE Practices, and economic and environmental performance of an organization. This is not unexpected because companies that adopt technology and CE practices usually perform well in economic and environmental areas.

### 4.2. Managerial Implications

A deliberate endeavor on the part of FMCG company administrations to seamlessly integrate technologies into their operational and logistical frameworks emerges as essential. This strategic move promises the dual advantage of reducing workforce requirements, thereby augmenting both economic and environmental performance. Furthermore, it serves as a pivotal step towards embracing digitalization and harnessing the potential of Artificial Intelligence, which is widely recognized as the future trajectory for the FMCG industry.

Simultaneously, significant attention must be directed towards embracing circular practices. An illustrative example includes the adoption of recycled materials for packaging purposes. The promotion and affordability of products adhering to circular norms could be facilitated by offering them at reduced prices.

These implications, stemming from the current study, carry policy implications for managerial decision-makers and stakeholders. The integration of technologies within Circular Economy practices to curtail carbon footprint while safeguarding financial stability has been endorsed by the study. Managers are encouraged to draw inspiration from this research, translating it into the implementation of technologies to reap operational, environmental, and financial benefits.

In the Indian context, governmental entities are urged to foster the adoption of technological infrastructure to support environmental legislation. This proactive stance aligns with the effective management of businesses in alignment with their ecological commitments. The study underscores the potential efficacy of providing businesses with tax exemp-

tions and interest-free loans as incentives for adeptly incorporating technology into their operations, a measure that could be envisaged by legislators based on the study's findings.

More recently, Industry 4.0 has had a positive impact on the environment [17]; hence, government should also design policies aimed at promoting industry 4.0.

## 5. Limitations and Future Directions of Research

The study's limitations suggest avenues for further investigation. Firstly, since data collection relied on an online survey questionnaire, future research could enhance generalizability by employing qualitative methods such as focus groups or interviews. Secondly, the study's conclusions are applicable specifically to the Indian context due to data from Indian Metro cities. To bolster reliability, future studies might gather data from various countries. Thirdly, due to data constraints, the study primarily addressed the economic and environmental aspects of sustainability, neglecting the social dimension. Subsequent research could delve into the social implications of technology adoption, examining effects on employment rates, inflation, and living standards within the FMCG sector. Lastly, while the current research employed SEM to test hypotheses, future studies could expand on this by incorporating additional CE practices and employing diverse mathematical and modeling tools. We also suggest future researchers to study green innovation, and green product and distribution channels and their impact on organizational performance.

## 6. Conclusions

The study focuses on improving economic and environmental performance within organizations by examining the impact of digitalization on Circular Economy (CE) practices. Data were gathered from Indian FMCG Companies using questionnaires across various communication channels. Structural Equation Modeling (SEM) was applied for analysis, yielding valid and consistent results. The findings show that digitalization positively affects CE practices and significantly enhances environmental performance. There is a strong connection between technological advancements and economic performance, while the influence of circular economy practices on economic performance is limited. The link between technological advancements and environmental performance is considered weak. Notably, technological advancements facilitated East European automakers in adopting CE practices effectively. CE practices bolster operational, economic, and environmental performance, supporting sustainability. The positive correlation between technological advancements, CE practices, and overall organizational performance is unsurprising, given their synergy. For FMCG leaders, integrating technology into operations is crucial for reducing workforce needs and improving performance. Embracing circular practices, such as using recycled materials, should also be prioritized. This has implications for decision-makers and stakeholders, advocating technology integration to reduce carbon footprint while maintaining stability. The study suggests governmental support for technology adoption in the Indian context, potentially through tax exemptions and interest-free loans to align policies with study insights.

**Funding:** This research received no external funding.

**Institutional Review Board Statement:** Not applicable.

**Informed Consent Statement:** Informed consent was obtained from all subjects involved in the study.

**Data Availability Statement:** Data is available on request.

**Conflicts of Interest:** The author declares no conflict of interest.

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
