# Peer review of "Economic and Environmental Efficiencies of Organizations: Role of Technological Advancements and Circular Economy Practices"

_sustainability, doi:10.3390/su152215935_

Round 1

Reviewer 1 Report

-        Page 2: The scientific question is not clear enough.

-        Page2: There is no flow in the text. It partly depends on the lack of proofreading but also on the fact that many statements and claims are made without being followed up by a clear and logical discussion. It is especially problematic in the Introduction that brings up a number of findings from different areas without linking them together.

-        Pages 1 and 2: In the introduction, you need to connect the state of the art to your paper goals. Please follow the literature review by a clear and concise state of the art analysis. This should clearly show the knowledge gaps identified and link them to your paper goals. Please reason both the novelty and the relevance of your paper goals.

-        Clearly discuss what the previous studies that you are referring to. What are the Research Gaps/Contributions? Please note that the paper may not be considered further without a clear research gap and novelty of the study.

-        Page3: Please add literature review section. Literature Review has the chance to be further improved: it seems that the authors have made the retrospection. However, via the review, what issues should be addressed? What is the current specific knowledge gap? What implication can be referred to? The above questions should be answered. Authors need to propose their study. A sustainable-circular citrus closed-loop supply chain configuration: Pareto-based algorithms. Journal of Environmental ManagementA New Wooden Supply Chain Model for Inventory Management Considering Environmental Pollution: A Genetic algorithm. Foundations of Computing and Decision Sciences Designing a sustainable closed-loop supply chain network considering lateral resupply and backup suppliers using fuzzy inference system. Environment, Development and Sustainability, please compare them with your paper in literature review section

-        Page3: Please explain your results into steps and links to your proposed method.

-        Page6: Please make sure your conclusions' section underscore the scientific value added of your paper, and/or the applicability of your findings/results, as indicated previously.

-        Page 13: Please revise your conclusion part into more details. Basically, you should enhance your contributions, limitations, underscore the scientific value added of your paper, and/or the applicability of your findings/results and future study in this session. The discussion is relatively simple and insufficient.

-        Page12: I recommend strengthening the comparison with previous research. Please compare the results in this study with those in previous studies. Discuss the study findings here. The discussion and conclusion are appropriately written and require no changes. The manuscript does not answer the following concerns: Why is it timeliness to explore such a study? What makes this study different from the previously published studies? Are there any similarly findings in line with the previously published studies?

-        Page 12:  Are the findings different from prior academic studies that were conducted elsewhere, if any?

-        Page13: I would like to request the author to emphasize the contributions practically and academically in the implication session.

There are also many formatting and grammar issues throughout the text. Modification traces are cleared.

Author Response

Dear Reviewer,

Thank you for your comments. Really we are grateful to you I have incorporated all your comments. 

Reviewer 2 Report

1) The abstract is not well written. It does not make sense in certain places. What does "The phenomenon of digitalisation commonly referred to as the bundle of novel technologies that aim to improve things constantly." mean? 

". Organisations must make advantage of emerging technology to ensure that operations are both economically and environmentally feasible" 

instead of "advantage of", organizations must aim at, is better wording. Further, What does emerging tech mean? Shouldn't it be environmentally friendly or green technologies? Logic error here. 

At line 17, the use of capital letters are unnecessary in places such as "...desire to measure the Economic and Environmental performance..."

Or, here for Approach:  SEM Approach should be just  SEM approach.

Annotations should be given for the reader (FMCG and SEM).

2) Why is the sample size adequate for SEM analysis? Isn't it adequate for the method? It could be argued that it could be more reliable if sample size is larger. Or, it also could be critisized that even for larger sample sizes, the representation quality and destribtional qualities are important, so, a large sample might not mean a good sample with representative capabilities for the population. Therefore, authors should include relevant statistics to show that data is appropriate. Report relevant statistics for this respect. Normality tests, skewness, kurtosis statistics should be reported and discussed with the concept above. 

3) Figure 1 shows the model. There are some weird arrows in my PDF version. Some of boxes are not visible. 

4) There are referencing errors both in terms of journals format and in terms of references already existent. Example is Khan et al 2021a and 2021. et al are not same authors. Also, there is 2021b in the text but not 2021a? Authors should carefully investigate and correct referencing errors. 

5)To what extent are the Indian FMCG companies that are included in the sample engage in circular economy practices? Why is the sector selected? It should be better discussed in abstract, introduction and the data part. Justification is needed. There is a relevant section dedicated for it in this paper. But it has no or very little contribution in terms of justifying how well and to what extent these companies engage in circular economy and in green tech innovation. Justification should be made with data from sector and table with relevant data to justify and to prove it. 

6) At line 55, sentence starts with "The second concept essential for gaining sustainability comes...". In the next paragraph, with "The Second and foremost objective of.." Where are the firsts? Why capital letter Second? 

7) Wording and Grammar errors in hypotheses. Example: 

H4: The Circular economy Practices influence positively to Environmental Performance.

8) What SEM is is given in line 253. It should be at the first page: "structural equation modelling (SEM)."

9) At table 9, capital, then no capital letters should be uniform. Relationship column should be the direction and also instead of ---- I suggest arrows. For H3 and H4, it is written as "accepted", but no asterices are given for estimates. 

10) Figures and Tables should have notes underneath. For example: for Table 9, what does * mean? For figure 3 what does bold or no-bold arrows mean? Statistical significance should be linked and explained here under the figures and tables. 

11) Discussion section should include discussion with existent empirical research. None exist. 

12) After discussion section, there has to be a Conclusion section. Move discussion above, take managarial implications and theoretical implications to the conclusion. Transform discussion into discussion and comparison with existent empirical literature. Then, extent the conclusion. 

13) Limitations and future directions are not well thought of. And, limitations section should be more compact. Future directions cannot be suggesting applying the model to other countries or taking effects on employment and inflation into consideration. This is a management paper with SEM method. Why not sustainability impacts, green innovation, green product and distribution channels are not discussed here but instead inflation? Why the climate change implications are not presented? 

At major level. 

Author Response

thank you for the remarks. 

Reviewer 3 Report

Issue 1.

The paper discusses the impact of digitalization and circular economy practices on the economic and environmental performance of Indian FMCG (Fast Moving Consumer Goods) companies. It highlights the increasing importance of digitalization in the Indian business scenario, particularly after the demonetization reform. The study examines how emerging technologies and sustainable practices can help companies become more economically and environmentally sustainable. The research analyzes data from 203 senior managers of FMCG companies using a SEM (Structural Equation Modeling) approach. The findings suggest that digital transformation and circular economy practices have a pervasive influence on both economic and environmental performance. The paper contributes to the literature by directly examining the relationship between technological advancements and economic and environmental performance, which has not been done in previous studies.

This research is important because it explores the impact of digitalization and circular economy practices on the economic and environmental performance of Indian FMCG companies, providing valuable insights for sustainable business strategies.

Issue 2.

An introduction section must be enlarged. Moreover, an enhanced literature review is required.

So, in the introduction section of the research paper, the following issues should be raised:

- The increasing significance of digitalization in the global business landscape and its implications for various industries, including FMCG.

- The specific context of the business scenario and the impact of the demonetization reform on digitalization and economic activities.

- The importance of sustainable practices and the circular economy concept in addressing environmental concerns and promoting long-term economic viability.

- The key examples of circular economy study in practice, such as mining and mineral resources management.

- The need for organizations, especially FMCG companies, to adopt emerging technologies and circular practices to enhance their economic and environmental performance.

- The research gap in understanding the direct relationship between technological advancements and economic and environmental performance, particularly in the Indian FMCG sector.

- The objectives of the study, which aim to measure the economic and environmental performance of Indian FMCG companies and explore how technology and circular practices influence these outcomes.

- The significance of the research findings in providing valuable insights for FMCG companies in formulating sustainable business strategies.

Issue 3.

In case to enhanced literature review according to Issue 2 please consider below mentioned papers.

Markevych, K.; Maistro, S.; Koval, V.; Paliukh, V. Mining sustainability and circular economy in the context of economic security in Ukraine. Min. Miner. Depos. 2022, 16, 101-113. https://doi.org/10.33271/mining16.01.101

Smol, M.; Marcinek, P.; Duda, J.; Szołdrowska, D. Importance of sustainable mineral resource management in implementing the circular economy (CE) model and the European green deal strategy. Resources, 2020, 9, 55. https://doi.org/10.3390/resources9050055

I believe it is worth considering in your paper.

Issue 4.

In-text references are mentioned incorrectly. It should be [1] instead of (Jabbour et al. 2018) and so on.

Issue 5.

Figure 1 is huge. Try to not use fonts in figures that is more than fonts in the paper.

Issue 6.

Figures 02 and 03 must be 2 and 3. Moreover, sharper quality is required.

Issue 7.

Reference must be prepared using MDPI formatting style.

Issue 8.

Based on a positive assessment of the article, taking into account the suggested revisions and considerations, it will be recommended that the article be published in the "Sustainability" journal.

Author Response

thank you for your valuable comments. 

Round 2

Reviewer 1 Report

No comment

Author Response

Thank you

Reviewer 2 Report

 Dear Authors,

1) One of the comments is not replied as it is visible in the rebuttal file.

Distributional qualities should be presented. Skewness and kurtosis are added in a table. By only these two, and without discussing the implications in light of these statistics, the critique is not well addressed. Main concern was, the identical independent distribution qualities of the sample. Not enough statistics are presented. Where is even the basic mean of dataset? Normality? See the relevant comment in the previous round.

2) Justification of company / sector selection: Another important issue that is not well addressed after the comments in the last round is, the justification of the selection of these companies. Why are they included? What are their committment to green manufacturing and other forms should be used to justify why authors chose this sectior. A table with comparative statistics should work and help on this justification and the sampling procedure section should be updated accordingly. 

3) Sustainability implications should be extended in the discussion. Following paper should be added and used to discuss the implications of industry 4.0 on the environmental sustainability. https://doi.org/10.1016/j.jclepro.2022.135786

4) Formating issues in the references section. 

5) English Grammar check is needed.

Minor problems such as typos or minor Grammar issues. 

Author Response

Response to Reviewer Feedback on Economic and Environmental Efficiencies of Organizations: Role of Technological Advancements and Circular Economy Practices

Dear Reviewer,

I hope this email finds you well. I wanted to express my sincere gratitude for taking the time to review my research paper titled on Economic and Environmental Efficiencies of Organizations: Role of Technological Advancements and Circular Economy Practices". Your insightful feedback and constructive comments have been invaluable in enhancing the quality of the paper.

I have carefully considered each of your comments and suggestions, and I'm pleased to inform you that I have made significant revisions to address the points you raised. Your expertise has guided me in refining the clarity and coherence of the paper, and I am confident that these changes will greatly improve its contribution to the field.

I have attached the revised version of the paper, with changes highlighted for your convenience. I would be highly appreciative if you could spare some time to review these revisions. If you have any further suggestions or concerns, please do not hesitate to share them. Your expertise is pivotal in helping me create a robust and impactful contribution to the field. I am committed to ensuring that the paper meets the highest standards, and your input is indispensable in achieving this goal.

Once again, thank you for your time, dedication, and insightful feedback. I look forward to hearing from you at your earliest convenience.

Thank you for your continued support and valuable contributions to the advancement of research in this area.

Warm regards,

Authors,

Comment 1. One of the comments is not replied as it is visible in the rebuttal file. Distributional qualities should be presented. Skewness and kurtosis are added in a table. By only these two, and without discussing the implications in light of these statistics, the critique is not well addressed. Main concern was, the identical independent distribution qualities of the sample. Not enough statistics are presented. Where is even the basic mean of dataset? Normality? See the relevant comment in the previous round.

Response: Agreed and done.

To assess normality of data, a z-test is applied as a normality test using skewness and kurtosis. A Z score could be obtained by dividing the skewness values or excess kurtosis value by their standard errors. For small sample size (n <50), z value ± 1.96 are sufficient to establish normality of the data. However, medium-sized samples (50≤ n <300), at absolute z-value ± 3.29, conclude the distribution of the sample is normal (Hair et al , 2013).  In this research Z statistic is greater than 3.50 for all research parameters ensuring normality of the data.

Construct

Mean

Std. Deviation

Z Value

TA

4.566

1.34

4.45

CP

5.434

1.08

3.89

EP

4.555

1.56

5.76

ECP

4.098

1.24

6.04

Comment 2: Justification of company / sector selection: Another important issue that is not well addressed after the comments in the last round is, the justification of the selection of these companies. Why are they included? What are their commitment to green manufacturing and other forms should be used to justify why authors chose this sector (tic). A table with comparative statistics should work and help on this justification and the sampling procedure section should be updated accordingly. 

Response: Agreed and done.

We have chosen FMCG sector because of following reasons:

The FMCG (Fast-Moving Consumer Goods) sector, the fourth-largest contributor to the Indian economy, is on a remarkable growth trajectory. As of December 2022, the FMCG market had already surged to a valuation of US$ 56.8 billion. Projections foresee a remarkable journey ahead, with the total revenue of the FMCG market poised to exhibit a robust Compound Annual Growth Rate (CAGR) of 27.9% between 2021 and 2027, potentially catapulting to nearly US$ 615.87 billion by 2027 (IBEF, 2023). What's particularly noteworthy is the sector's resolute commitment to sustainability and circular economy practices. A noteworthy 60% of FMCG companies in India have embraced at least one circular economy practice, as highlighted in the FICCI Report of 2022. This dedication extends further, as a survey by the Confederation of Indian Industry (CII) revealed that an impressive 75% of FMCG companies operating in India are actively engaged in sustainability initiatives (CII, 2023).

Aligned with these sustainability objectives, the Indian government has set an ambitious target to reduce waste generation by a substantial 30% by the year 2030 (IBEF, 2023). To foster green manufacturing, a series of regulations have been introduced, including the Energy Conservation Act and the Water Conservation Act. FMCG companies are responding to these imperatives by integrating sustainable ingredients and materials into their product portfolios. For instance, Hindustan Unilever has committed to incorporating 100% sustainable palm oil in its products by 2023, while Nestle India is diligently working towards using 100% recycled paper in its packaging by 2025 and enhancing packaging recyclability. The tangible expression of this commitment can be observed in Coca-Cola India's innovative introduction of a bottle crafted entirely from 100% recycled plastic and PepsiCo India's launch of snacks thoughtfully packaged in compostable wrappers. Moreover, these companies are proactively conserving water within their operations; Dabur India, for instance, has deployed water-saving devices in its manufacturing facilities, and Godrej Consumer Products has introduced a rainwater harvesting system at its corporate headquarters.

Beyond this, significant strides are being made in enhancing energy efficiency, as exemplified by ITC's installation of solar panels at its factories and Marico's implementation of energy-efficient lighting within its offices. Additionally, FMCG companies are placing an emphasis on responsible waste management to minimize their environmental footprint. Notable initiatives include Procter & Gamble India's establishment of a waste-to-energy plant at its Manesar factory and HUL's pioneering recycling program for its shampoo bottles.

These noteworthy sustainability milestones underscore the substantial progress achieved by FMCG companies in India in their unwavering commitment to sustainability. Therefore, the choice of the FMCG sector as the focal point of our study is well-justified, as it embodies the transformative journey toward sustainability and green manufacturing.

Comment 3: Sustainability implications should be extended in the discussion. Following paper should be added and used to discuss the implications of industry 4.0 on the environmental sustainability. https://doi.org/10.1016/j.jclepro.2022.135786

Response: Agreed and done. We have extended the discussion section after reading the following paper that truly has advanced our understanding.

Bildirici, M., & Ersin, Ö. Ö. (2023). Nexus between Industry 4.0 and environmental sustainability: A Fourier panel bootstrap cointegration and causality analysis. Journal of Cleaner Production, 386, 135786

Comment 4: Formating issues in the references section. 

Response: We wholeheartedly concur with this evident point; however, to adhere to the prescribed guidelines for authors on the sustainability journal's website, we have chosen to adopt the free format style. For further reference, kindly follow the provided link: https://www.mdpi.com/journal/sustainability/instructions 

Comment 5: English Grammar check is needed.

Response:  Agreed and done.

Reviewer 3 Report

Dear authors, I wish to express my utmost satisfaction with the corrections you have implemented. My congratulations to you for your diligent work.

Author Response

Thank you

Round 3

Reviewer 2 Report

Dear Authors, critiques responses are satisfactory. My decision is positive for this version. 

No problems.